# Prevalence of floating toe and its relationship with static postural stability in children: The Yamanashi adjunct study of the Japan Environment and Children's Study (JECS-Y)

Taro Fujimaki[1], Masanori Wako[1]*, Kensuke Koyama[1], Naoto Furuya[1], Ryoji Shinohara[2], Sanae Otawa[2], Anna Kobayashi[2], Sayaka Horiuchi[2], Megumi Kushima[2], Zentaro Yamagata[2], Hirotaka Haro[1], on behalf of The Yamanashi adjunct study of the Japan Environment and Children's Study Group[¶]

1 Department of Orthopaedic Surgery, Faculty of Medicine, University of Yamanashi, Chuo-shi, Yamanashi, Japan, 2 Center for Birth Cohort Studies, Interdisciplinary Graduate School of Medicine, University of Yamanashi, Chuo-shi, Yamanashi, Japan

¶ Membership of The Yamanashi adjunct study of the Japan Environment and Children's Study Group is provided in the Acknowledgments.
* wako@yamanashi.ac.jp

**Data Availability Statement:** All relevant data are within the paper and its Supporting Information files.

## Abstract

Floating toe (FT) is a frequently seen condition in which a toe is inadequately in contact with the ground. Although toes play an important role in stabilizing standing posture and walking, many aspects of the effects of FT on the body remain unclear. To our knowledge, there have been no reports about the relationship between FT and postural stability, especially in children. This study aimed to clarify the prevalence of FT and its relationship with static postural stability in children. Of the 400 children aged 8 years who participated in our cohort study, 396, who were examined for static postural stability, were included in this study. Postural stability and FT were assessed using a foot pressure plate. The sway path length of the center of pressure and the area of the ellipse defined as the size of the area marked by the center of pressure, were measured as an evaluation of static postural stability. We calculated the "floating toe score (FT score: small FT score indicates insufficient ground contact of the toes)" using the image of the plantar footprint obtained at the postural stability measurement. The rate of FT was elevated at more than 90%, and the FT score in the eyes-closed condition was significantly higher than that in the eyes-open condition in both sexes. The FT score significantly correlated with the center of pressure path and area. Our results suggest that ground contact of the toes is not directly related to static postural stability in children, but it may function to stabilize the body when the condition becomes unstable, such as when the eyes are closed.

## Introduction

Human feet support bodyweight, absorb impact, and push the body forward while walking, and the forefeet play an important role in standing firmly on the ground, stabilizing the body,

**Funding:** The authors received no specific funding for this work.

**Competing interests:** The authors have declared that no competing interests exist.

and walking and running [1, 2]. The toes are in contact with the ground for approximately three-quarters of the stance phase during walking and they distribute the load [3]. Toes are also thought to play an important role in the ability to stand firmly on the ground by stabilizing the body [4]. Therefore, toe function is important for preserving healthy daily activities such as standing, moving, and walking.

Recently, "floating toe" (FT) has received attention as a possible cause of toe dysfunction [4, 5]. Originally, the condition reportedly occurred as a result of surgery, and is one of the most common complications of Weil osteotomy [6, 7]. Previous studies concluded that FT results from excessive dorsiflexion or a lack of plantarflexion of the metatarsophalangeal joints [8–10]. Studies in Japan reported that FT influences dynamic balance, stride length, and walking speed [4, 11]. Fukuyama et al. defined FT as a condition in which the toes do not contact the ground in the standing position and the weight does not shift to the toe while walking.

Although there are many unclear aspects of the effects of FT on the body in children, it is speculated that FT has some relation to body stability if the condition is due to functional deterioration of the toes. However, there are no reports on the relationship between FT and postural stability, and it is not clear whether FT itself is an adverse condition in children.

Our institution has been conducting a cohort study of 8-year-old children since 2019, and we have been measuring the plantar pressure and static postural stability in the participants of this cohort study. Hence, we could meet the purpose of this study, which was to clarify the prevalence of FT and its relationship with static postural stability in 8-year-old children in this cohort.

## Materials and methods

### Study design

The Japan Environment and Children's Study (JECS), which is a national project funded directly by the Ministry of the Environment, Japan, is a birth cohort study undertaken to elucidate the influence of environmental factors during the fetal period and early childhood on children's health, with follow-up until age 13. Details of the protocol and baseline data of the JECS are available elsewhere [12]. In our institution, additional survey is being performed for children who will be 8 years among JECS participants. This additional survey includes ophthalmologic or oral investigations and postural stability test using the foot pressure plate. The ethics committee of the School of Medicine, University of Yamanashi approved this additional survey (approval number: 2020). Written informed consent was obtained from all participants' mothers or their partners in accordance with the Declaration of Helsinki.

### Participants

Of the 400 children aged 8 years who participated in this additional survey conducted at our institution between July 2019 and February 2020, 396 children who were examined for static postural stability were included in the study. Four cases were excluded because it was impossible to measure static postural stability according to the protocol due to restlessness.

### Test procedure and protocol

Body height was measured and recorded in centimeters to the nearest millimeter; body weight was measured to the nearest 0.1 kg using an electronic weighing scale, with the participant wearing shorts and a T-shirt. The Rohrer index was calculated using the following formula: Rohrer index = $10 \times$ height (m)/weight (kg)$^3$.

Static postural stability and FT were assessed using a foot pressure plate (Win-Pod, Medi-capteurs, France). All participants were instructed to maintain an upright standing position on the platform, barefoot, with their arms hanging by their sides and their feet parallel to each other. They were tested two times with their eyes open and two times with their eyes closed, each test lasting 20 s. The inspection was performed using the postural mode. Of the several parameters for static postural stability that can be measured with the foot pressure plate, we evaluated the following two measurements, which are more common: the total sway path length of the center of pressure (COP-path) and the area of the ellipse (COP-area):

- COP-path [mm]–defines the total length of the path marked by the COP; the sum of distances between the locations of the COP constitutes the path length.

- COP-area [mm$^2$]–defines the size of the area marked by the COP; ellipse area includes 95% of the COP measurement points; this parameter makes it possible to assess the size of the area of the COP movement on the support surface.

Of the two measurements, one with the smaller COP-path was used for the analysis.

Based on a previous report [4], we calculated the floating toe score (FT score) using the image of the plantar footprint obtained at the postural stability measurement. As with postural stability, the footprint with the smaller COP-path was used. For the 10 toes of both feet, if a toe appeared clearly on the image (shown in red to green in the plantar pressure chart), 2 points were given; if it appeared unclearly (shown in blue in the plantar pressure chart), 1 point was given; and if it did not appear, no points were given. The points were summed to realize the FT score (Fig 1 shows an example of a plantar pressure chart). If FT score was ≥18 points and the big toe of both feet had gained 2 points, the participants were placed in the "contact toe" group; those with 11 to 17 points were placed in the "incomplete contact toe" group, and those with ≤10 points were placed in the FT group.

## Statistical analyses

To evaluate the intraobserver agreement for FT score, the measurements of 20 randomly selected plantar footprints were repeated by the same reader (T.F.) during the course of two sessions at least 1 month apart. For interobserver agreement, a second reader (M.W.) repeated the measurements for the same 20 participants. Interobserver and intraobserver reliabilities

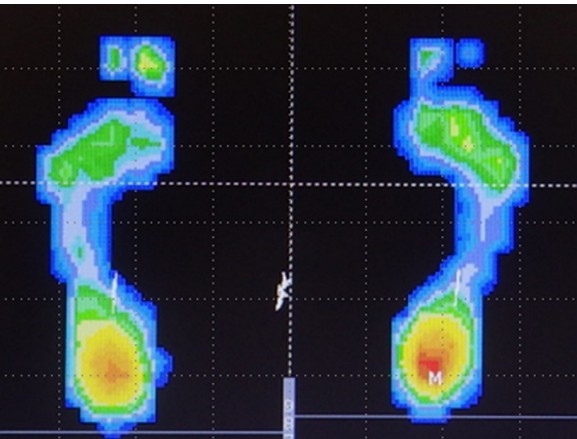

**Fig 1. Example of plantar pressure chart.** One yellow and one green toe on the right foot and one yellow and one blue toe on the left foot resulted in an FT score of 7 points in this case.

for FT score were assessed by estimating intraclass correlation coefficients (ICCs) along with 95% confidence intervals (CIs) using an ICC (2,1) modeling scheme.

The unpaired t-test was used for investigating the sex differences of each parameter. The paired t-test was used to examine the differences in FT score, COP-path, and COP-area between the eyes-open and eyes-closed conditions. Pearson's correlation coefficient was used to investigate the correlations between each measurement. Statistical significance was set at p <0.05.

## Results

The interobserver and intraobserver reliabilities of FT score of the 20 randomly selected participants were 0.969 (95% CI, 0.924–0.988) and 0.989 (95% CI, 0.973–0.996), respectively. These values indicated substantial agreement (ICC, >0.9).

Table 1 shows the summary of height, weight, and Rohrer index of all participants. There were no significant gender differences in height, weight, and Rohrer index, and none of these participants had an extreme body posture.

Table 2 shows the results of postural static stability. COP-path and COP-area of the total, female, and male participants in the eyes-closed condition were significantly larger than those in the eyes-open condition. Significant differences between boys and girls were observed in the COP-path and COP-area in eyes-closed conditions.

Table 3 shows the results of FT score. According to Fukuyama et al.'s criteria [4], the rate of FT was very high at more than 90% under all conditions. FT score in the eyes-closed condition was significantly higher than that in the eyes-open condition in both sexes. There were no significant gender differences in the FT score.

Table 4 shows the correlations between static postural stability and FT score in the eyes-open and eyes-closed conditions. FT score had a significantly moderate correlation with COP-path and COP-area in both eyes-open and eyes-closed conditions in boys and significant but weak correlation in girls.

## Discussion

We assessed 396 participants who were 8-year-old children for FT and static postural stability. COP-path and COP-area in the eyes-closed condition were significantly larger than those in the eyes-open condition, and the postural stability in girls was higher than that in boys in the eyes-closed condition. We found a fairly high rate of FT in all participants and a higher FT score in the eyes-closed condition than in the eyes-open condition. Moreover, there were significant correlations between the FT score and COP-path and COP-area. To our knowledge, this is the first report on the relationship between FT and postural stability.

The result suggesting higher static postural stability in females is similar to that in previous reports. A majority of previous studies have found that the balancing ability of girls is better than that of boys and that the sex differences in postural stability among children may explain maturational differences in the central nervous structures [13–18]. de Sá et al. reported that in children, the visual system matures before the vestibular system; therefore, the open-eyes

**Table 1. Height, weight, and Rohrer index of all participants (mean ± SD).**

|  | total (n = 396) | female (n = 216) | male (n = 180) |
|---|---|---|---|
| height (cm) | 124.8±5.0 | 125.0±4.8 | 124.6±5.1 |
| weight (Kg) | 24.7±4.3 | 24.8±4.2 | 24.5±4.5 |
| Rohrer index | 126.2±14.2 | 126.4±14.8 | 126.0±13.6 |

**Table 2. COP-path and COP-area (mean ± SD).**

|  | condition | total (n = 396) | female (n = 216) | male (n = 180) |
|---|---|---|---|---|
| COP-path | EO | 200.1±97.5 | 191.7±81.5 | 210.3±113.2 |
|  | EC | 291.3±147.2* | 274.1±135.4* | 312.0±158.2*† |
| COP-area | EO | 192.5±162.7 | 182.4±142.1 | 204.8±184.1 |
|  | EC | 320.7±278.0* | 285.1±223.5* | 363.3±327.3*† |

EO = eye open, EC = eye closed, COP-path = the total displacement of center of pressure. COP area = the area of the mean center of pressure.

*: significantly different with EO (p < 0.05, paired t-test).

†: significantly different with female (p < 0.05, unpaired t-test).

postural stability is first achieved at 5 to 7 years of age before the closed-eyes postural stability [13]. The vestibular system is believed to mature faster in girls. Hirabayashi et al. showed that girls were superior to boys with respect to vestibular function at the age of 7–8 years [14]. Lenroot et al. reported that girls reached peak values of brain volumes earlier than boys [15]. The current study revealed that the static postural stability of girls is better than that of boys only in the eyes-closed condition. These results may be due to the dominance of the vestibular system in using vestibular cues under the condition of no visual cues and inaccurate somatosensory input. Thus, the results of static postural stability are almost the same as those in previous reports.

The toe plays an important role in stabilizing the standing posture and walking by increasing the ground contact area [2, 3], and FT is a condition in which the toes do not contact the ground in the standing position. In recent years, some reports have shown that the frequency of FT in children ranges from 40% to 98%. Araki et al. assessed 198 children aged 3 to 5 years using footprint images and reported that FT was found in 87.7% to 98.7% of them [5]. Tasaka et al. studied 635 children aged 9 to 11 years and reported that 40.3% of all feet had no toe contact with the floor at all, and they were concerned about the trend of declining foot function in children [10]. Despite differences in the methods used by each author to assess FT, the rate of FT was similarly high in the current study. Although there are some reports that FT is due to toe dysfunction and it is a pathological condition [4, 11], we believe that FT in children has little pathological significance because it is very common at least at 8 years of age.

Although there have been some reports on postural stability and foot posture, there has been no English report on the relationship between postural stability and FT. The current study showed that the body was more unstable in cases with more ground contact toes. If toe contact is directly important for postural stability, the greater the FT score, the more stable will be the center of gravity. The results of the current study indicate that the larger the FT score, the greater the COP-path and COP-area, suggesting that toes stabilize the body that becomes unstable when eyes are closed. In other words, ground contact of the toes is not directly related

**Table 3. Floating toe score (mean ± SD) and classification.**

|  |  | female (n = 216) | | | male (n = 180) | | |
|---|---|---|---|---|---|---|---|
|  |  | EO | EC | p-value | EO | EC | p-value |
| FT score |  | 3.6±2.4 | 4.4±3.1* | 0.000 | 3.7±3.3 | 4.9±3.7* | 0.000 |
| classification n (%) | FT | 211 (97.7) | 205 (94.9) |  | 172 (95.6) | 165 (91.7) |  |
|  | incomplete | 5 (2.3) | 10 (4.6) |  | 6 (3.3) | 13 (7.2) |  |
|  | contact toe | 0 (0) | 1 (0.5) |  | 2 (1.1) | 2 (1.1) |  |

EO = eye open, EC = eye closed, FT = floating toe.

**Table 4. Pearson's correlation coefficient of the measurements.**

| | FT score | | | |
|---|---|---|---|---|
| | **female** | **p-value** | **male** | **p-value** |
| **a: eyes-open condition** | | | | |
| COP-path | 0.275 | 0.000 | 0.495 | 0.000 |
| COP-area | 0.220 | 0.001 | 0.480 | 0.000 |
| **b: eyes-closed condition** | | | | |
| COP-path | 0.411 | 0.000 | 0.545 | 0.000 |
| COP-area | 0.352 | 0.000 | 0.578 | 0.000 |

EO = eye open, EC = eye closed, FT = floating toe.

COP-path = the total displacement of center of pressure. COP-area = the area of the mean center of pressure.

to static postural stability in children, but it may function to stabilize the body when the condition becomes unstable. Moreover, the current study revealed that the FT score of the total, female, and male cases in the eyes-closed condition was greater than that in the eyes-open condition (FT is more frequent in eye-closed condition), and there is no similar report in the past. This is probably the result of grounding the toes in an attempt to control the unstable body caused by eyes closure and may support the theory described above. In the future, the evaluation of FT in unstable or dynamic situations will clarify the significance of FT.

Our study had several limitations. First, we evaluated FT using the plantar pressure diagram obtained from the foot pressure plate. As there is no standard method to evaluate FT, it is not exactly possible to compare the results of the current study with previous reports on the frequency of FT. However, the number of cases is sufficient in our study, and we think there is no doubt about the results of the high frequency of FT. Second, in the present study, only the interrelationship between FT and static postural stability was examined. Based on our results indicating lesser static postural stability in cases with higher FT scores, we found no direct relationship between FT and static postural stability. However, we were not able to prove it directly. We speculate that various other factors are involved among these factors in a complex manner. Furthermore, it has been reported that static postural stability reflects several physical factors other than nervous system maturation. Angin et al. reported that postural sway velocity increases with pronation of the foot [19]. Likewise, Cote et al. reported that postural stability was greater in pronators than in supinators [20]. In the current research series, we have measured and saved data on plantar footprints, physical exercise habits of individuals and their parents, blood investigations, body composition such as body fat and muscle mass, and Pediatric Evaluation of Disability Inventory–Computer Adaptive Test (PEDI-CAT) to assess their mental development. In the future, we plan to investigate FT and postural stability in children using these data in a more multifaceted way. Moreover, we also plan to follow-up with the same participants in this cohort and perform similar tests, which will allow us to assess changes in FT and postural stability over time.

As a side note, this study was conducted as an additional study to the Ministry of the Environment's JECS. The views expressed in this paper are the authors' own and not those of the Ministry of the Environment.

In conclusion, this study demonstrated that the frequency of FT in healthy 8-year-old children was very high. Our results suggested that FT is not directly related to retention of the standing posture in children; however, the toes do play a role by making ground contact in conditions when static postural stability is compromised and the standing posture becomes unstable. At least at 8 years of age, although FT is very common, it is not directly related to postural control and considered to have minor pathological significance.

## Supporting information

**S1 Data.**
(XLSX)

## Acknowledgments

We are grateful to all the participants of the JECS-Y and to all individuals involved in data collection. We also thank the following members of the JECS-Y as of 2020: Zentaro Yamagata (Principal investigator, e-mail: zenymgt@yamanashi.ac.jp), Ryoji Shinohara, Sanae Otawa, Anna Kobayashi, Sayaka Horiuchi, and Megumi Kushima (Center for Birth Cohort Studies, Interdisciplinary Graduate School of medicine, University of Yamanashi, Chuo, Japan); Takeshi Inukai and Emi Sawanobori (Department of Pediatrics, School of Medicine, University of Yamanashi, Chuo, Japan); Kyoichiro Tsuchiya (Third Department of Internal Medicine, University of Yamanashi, Chuo, Japan); Takahiko Mitsui (Department of Urology, Interdisciplinary Graduate School of Medicine, University of Yamanashi, Chuo, Japan); Kenji Kashiwagi (Department of Ophthalmology, University of Yamanashi, Chuo, Japan); Daijyu Sakurai and Hiroyuki Watanabe (Department of Otorhinolaryngology-Head and Neck Surgery, School of Medicine, University of Yamanashi, Chuo, Japan); Koichiro Ueki and Naana Baba, (Department of Oral and Maxillofacial Surgery, Interdisciplinary Graduate School of Medicine, University of Yamanashi, Chuo, Japan); and Hiroshi Yokomichi, Kunio Miyake, Yuka Akiyama, Tadao Ooka, and Reiji Kojima (Department of Health Sciences, School of Medicine, University of Yamanashi, Chuo, Japan).

## Author Contributions

**Conceptualization:** Masanori Wako.

**Formal analysis:** Masanori Wako, Ryoji Shinohara.

**Investigation:** Taro Fujimaki, Masanori Wako.

**Project administration:** Ryoji Shinohara, Sanae Otawa, Sayaka Horiuchi, Megumi Kushima, Zentaro Yamagata.

**Supervision:** Zentaro Yamagata, Hirotaka Haro.

**Writing – original draft:** Masanori Wako.

**Writing – review & editing:** Taro Fujimaki, Kensuke Koyama, Naoto Furuya, Anna Kobayashi, Zentaro Yamagata, Hirotaka Haro.

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
