## [Decision Letter · Decision Letter 0]

2 Feb 2021

PONE-D-21-00839

Prevalence of floating toe and its relationship with static postural stability in children: The Yamanashi adjunct study of the Japan Environment and Children’s Study (JECS-Y)

PLOS ONE

Dear Dr. Wako,

Thank you for submitting your manuscript to PLOS ONE. After careful consideration, we feel that it has merit but does not fully meet PLOS ONE’s publication criteria as it currently stands. Therefore, we invite you to submit a revised version of the manuscript that addresses the points raised during the review process.

We look forward to receiving your revised manuscript.

Kind regards,

Tomoyoshi Komiyama, Ph.D

Academic Editor

PLOS ONE

Additional Editor Comments:

Dear Authors,

The authors results showed that FT is not directly related to retention of the standing posture in 396, 8-year-old children. The authors found a fairly high rate of FT in all participants and a higher FT score in the eyes-closed condition than in the eyes-open condition. Moreover, there were significant correlations between the FT score and the COP-path and COP-area. This is the first report on the relationship between FT and postural stability.

If these results become clear, I think your research is important for the future of clinicians and clinical researchers who aim to better understand FT.

However, I think that it is necessary to strengthen the reliability of the result by adding as much information as possible.

For that reason, I have some questions and concerns about the manuscript that you might consider.

I have attached the comments that reviewers have pointed out in your manuscript.

Two reviewer are critical to your manuscript.　

I think that the points suggested by these reviewers will be helpful in the revision.

In addition, I read your manuscript.

1. I think your research is good, but your methods are not clear enough for most readers to understand.

I strongly recommend the authors check and correct their manuscript.

2. I would like you to clearly describe how the clarification of this research will be useful in clinical practice.

Journal Requirements:

2. Thank you for including your ethics statement:  "This study was approved by the institutional review board of our university (No 2020)."

3. Please include your actual numerical p-values in Tables 3 and 4.

4. In your Methods section, please provide additional information about the participant recruitment method and the demographic details of your participants. Please ensure you have provided sufficient details to replicate the analyses such as:

a) a table of relevant demographic details,

b) a statement as to whether your sample can be considered representative of a larger population, and

c) a description of how participants were recruited.

5. Please provide a citation for The Japan Environment and Children’s Study.

6. One of the noted authors is a group or consortium [Japan Environment and Children’s Study

9 Group]. In addition to naming the author group, and listing the individual authors and affiliations within this group in the acknowledgments section of your manuscript, please also indicate clearly a lead author for this group along with a contact email address.

Reviewers' comments:

Reviewer's Responses to Questions

**Comments to the Author**

1. Is the manuscript technically sound, and do the data support the conclusions?

Reviewer #1: Partly

Reviewer #2: Partly

2. Has the statistical analysis been performed appropriately and rigorously? 

Reviewer #1: Yes

Reviewer #2: Yes

3. Have the authors made all data underlying the findings in their manuscript fully available?

Reviewer #1: Yes

Reviewer #2: Yes

4. Is the manuscript presented in an intelligible fashion and written in standard English?

Reviewer #1: Yes

Reviewer #2: Yes

5. Review Comments to the Author

Reviewer #1: This paper aims to clarify the prevalence of floating toe and its relationship with static postural stability in children. An objective evaluation is attempted by using a foot pressure plate. In conclusion, it was shown that the finger's contact with the ground is not directly related to postural stability but is more stable when the eyes are closed.

The following items are raised as questions in this paper's peer review, and the authors should clarify these questions.

For "Study design," why did the authors target only at the age of 8 in this study?

For "Test procedure and protocol", why did you compare COP-path and COP-area and use the measured value with the smaller COP-path for the analysis?

The authors cite J Phys Ther Sci, but the subject used here is an adult, and if it is adapted to children, it may be necessary to re-evaluate the parameters.

As for the FT score, the visual part is large, and it is necessary to show the definition when counting points.

Regarding the results, the values of men and women summed in Table 3 and Table 4, but what is the meaning of adding the significance of different qualities in the first place? If the total value is to be compared, it may be limited to the case of comparing all ages or comparing over time.

Regarding the discussion, the authors mentioned the difference in the central nervous system structure regarding the difference between men and women (line170-173). Still, it was explicitly caused by the difference between men and women in this study's central nervous system. It should be clarified whether or not it matches the data.

Line 173-182 suggests that the visual system precedes the vestibular system. The optical system is blocked when the eyes are closed, suggesting that the vestibular system's superiority over boys affects girls.

Neurologically, the static postural stability is controlled by the sensory nerve tract, the sensorimotor integration center, and the motor nerve tract.　In this respect, if it results only in the visual system and vestibular system pointed out by the authors, some scientific verification is necessary.

As mentioned above, at least the subjects should be targeted at a wider age group, and the research method should be reviewed, and the data should be interpreted again, such as improving the objectivity of the FT score and adding neuroscience experts to the research organization.

Reviewer #2: In this study, the relationship between floating toe and static postural stability in children is investigated through experiments.

Although it can be evaluated as a valuable achievement for many subjects, there are some parts that are difficult to understand and some parts that require additional explanation, so they are listed below.

I would like you to answer them appropriately.

If there are any misunderstandings or mistakes due to my lack of knowledge, please forgive me and report it as an answer to me.

1. At the end of the Abstract and in the Discussion section, the following are concluded: The results of this study show that contact between the toes and the ground is not directly related to static postural stability, but contributes to body stability during postural instability.

How will this result be used for future research and medical care?

2. Please correct the explanation of COP-path and COP-area so that the reader can understand it intuitively by using the example.

Also, please add the reason why the measurement time of 20 seconds was decided.

The reason why the shorter COP-path was used for the analysis in the two experiments is also unclear as it is.

Furthermore, I don't know if two experiments are enough.

Please add descriptions.

3. I can understand the definition of the explanation of FT score used in this study, but I do not understand its specific meaning, for example, why the thumb touches the ground at 2 points.

Please indicate the reason for deriving this definition.

In addition, it is said that there are other FT evaluation indexes, so please explain the relationship with them and the original points that are significantly different.

In the text, there are some descriptions such as FT rate and FTS.

Make sure it's different or that it's the same thing with a different name.

4. In the discussion, you said that the greater correlation between the FT score and the COP-path or COP-area when the eyes are closed is a result of the toes performing a function to stabilize the body.

But can't we interpret that instability is the cause of the toes' contact with the ground?

I think this is an important point, so please add a description.

5. "prevalence of floating toe" is included in the title.

Did you get any new insights into this?

In the discussion, it was written as the same as the previous reports, and I could not find a description of what was newly discovered.

6. It is very interesting that you are planning multifaceted research in the future, so please tell us specifically what kind of research results you can expect.

6. PLOS authors have the option to publish the peer review history of their article (what does this mean?). If published, this will include your full peer review and any attached files.

Reviewer #1: No

Reviewer #2: No

---

## [Author Response · Author response to Decision Letter 0]

25 Feb 2021

Response to Editor:

Comment 1:

I think your research is good, but your methods are not clear enough for most readers to understand.

I strongly recommend the authors check and correct their manuscript.

Response: 

Thank you for your suggestion. The methods section has been revised based on the reviewers’ comments.

Comment 2:

I would like you to clearly describe how the clarification of this research will be useful in clinical practice.

Response: 

Due to the importance of the toes in locomotion, floating toe that is not grounded is considered a bad condition for the human body. In the past, it has been reported that the dynamic balance of adults with floating toe is reduced, but the true significance of floating toe in children is not well understood (reference No 4, 11).

The results of the current study showed that floating toe is very common in children. This would imply that floating toe in 8-year-old children is not a morbid condition to worry about.

And in this study, it was found that grounding of the toes can compensate for standing instability. But FT in a dynamic condition is still not well understood. In the future, the evaluation of FT in unstable or dynamic situations will clarify the significance of FT.

I have added these points to the manuscript. (Line 199-201, 210-215)

Response to Journal Requirements:

Comment 1:

Response: 

I have revised the study design section according to your comment. (Lines 73-75)

Comment 2:

Please include your actual numerical p-values in Tables 3 and 4.

Response: 

Thank you for your suggestion. I have revised Tables 3 and 4 based on the comments of you and reviewer #1.

Comment 3:

In your Methods section, please provide additional information about the participant recruitment method and the demographic details of your participants. Please ensure you have provided sufficient details to replicate the analyses such as:

a) a table of relevant demographic details,

b) a statement as to whether your sample can be considered representative of a larger population, and

c) a description of how participants were recruited.

Response: 

Thank you for your suggestion. I have revised the manuscript regarding the registration process for participants.

Comment 4:

Please provide a citation for The Japan Environment and Children’s Study.

Response:

　We agree with the importance of the citation. However, we already have cited it as reference No 12.

Comment 5:

One of the noted authors is a group or consortium [Japan Environment and Children’s Study Group]. In addition to naming the author group, and listing the individual authors and affiliations within this group in the acknowledgments section of your manuscript, please also indicate clearly a lead author for this group along with a contact email address.

Response:

This study was conducted by the Yamanashi adjunct study of the Japan Environment and Children’s Study (JECS-Y) group. The principal investigator of the group is Zentaro Yamagata. I have added this information at acknowledgments. (Lines 247-248)

Response to Reviewer #1: 

Comment 1:

For "Study design," why did the authors target only at the age of 8 in this study?

Response:

Thank you for your comment. Among the JECS participants, the volunteers who will be 8 years old in 2019 were surveyed with a variety of additional surveys unique to our university, including ophthalmologic or oral surveys or postural stability tests in addition to the nationally standardized survey. Therefore, although the study was conducted on 8-year-olds, our study participants were not recruited only for the tests of postural stability and foot condition. Please see the study design section, which has been revised. (Lines 71-75) 

Comment 2:

For "Test procedure and protocol", why did you compare COP-path and COP-area and use the measured value with the smaller COP-path for the analysis? The authors cite J Phys Ther Sci, but the subject used here is an adult, and if it is adapted to children, it may be necessary to re-evaluate the parameters.

Response:

Thank you for your question. In general, there are several parameters for evaluating postural stability. Also, the foot pressure plate used in this study can measure multiple parameters such as the velocity of the center of gravity or the trajectory length per unit area in addition to the total sway path or the area of the ellipse. We believe that the detailed significance of each parameter and the differences between them have not yet been clarified. Therefore, we selected two of the more common parameters.

The test procedure and protocol section for postural stability evaluation has been revised to make it easier to understand.

Some of the literature that measured postural stability using force plate discuss the results using the average of the two-times measurements, but most of the literature uses the one with the better results (the smaller value of postural stability). In particular, in the case of children, I think it makes more sense to use the one with less sway because the variation of postural stability in each measurement is larger due to the effects of concentration and other factors.

Comment 3:

As for the FT score, the visual part is large, and it is necessary to show the definition when counting points.

Response:

Thank you for your suggestion. I had a similar comment from the other reviewer. 

I have revised the test procedure and protocol section. (Line 100-110)

Comment 4:

Regarding the results, the values of men and women summed in Table 3 and Table 4, but what is the meaning of adding the significance of different qualities in the first place? If the total value is to be compared, it may be limited to the case of comparing all ages or comparing over time.

Response:

I agree with your assessment. I think that the total value is unnecessary. Therefore, I have revised the tables.

Comment 5:

Regarding the discussion, the authors mentioned the difference in the central nervous system structure regarding the difference between men and women (line170-173). Still, it was explicitly caused by the difference between men and women in this study's central nervous system. It should be clarified whether or not it matches the data.

Response:

Thank you for your suggestion. I agree with your opinion, but this is clearly stated in lines 186-190.

Comment 6:

Line 173-182 suggests that the visual system precedes the vestibular system. The optical system is blocked when the eyes are closed, suggesting that the vestibular system's superiority over boys affects girls.

Neurologically, the static postural stability is controlled by the sensory nerve tract, the sensorimotor integration center, and the motor nerve tract.　In this respect, if it results only in the visual system and vestibular system pointed out by the authors, some scientific verification is necessary.

As mentioned above, at least the subjects should be targeted at a wider age group, and the research method should be reviewed, and the data should be interpreted again, such as improving the objectivity of the FT score and adding neuroscience experts to the research organization.

Response:

Thank you for your comment. The main outcome showed in this study is that when the standing position becomes unstable due to closed eyes, the toes become more functional and grounded.

Since the result that 8-year-old girls have higher postural stability with closed eyes is not the main outcome of this study, I have only discussed that this result is as same as the previous reports that girls have higher postural stability with closed eyes at the age of 7 to 10 years because girls’ vestibular system matures earlier than that of boys.

As you pointed out, various factors other than the visual and vestibular systems are involved in the postural control. I agree with the necessity to investigate a wider range of age groups from various perspectives to estimate the detailed mechanism of postural control. I believe that this is one of our future tasks. We are also planning to conduct an additional survey for a similar cohort in the future, so we would like to perform the same tests and compare the results with those of this study. I have revised the discussion section based on these points.

Response to Reviewer #2

Comment 1:

At the end of the Abstract and in the Discussion section, the following are concluded: The results of this study show that contact between the toes and the ground is not directly related to static postural stability, but contributes to body stability during postural instability.

How will this result be used for future research and medical care?

Response:

Due to the importance of the toes in locomotion, floating toe that is not grounded is considered a bad condition for the human body. In the past, it has been reported that the dynamic balance of adults with floating toe is reduced, but the true significance of floating toe in children is not well understood.

The results of the current study showed that floating toe is very common in children. This would imply that floating toe in 8-year-old children is not a morbid condition to worry about.

Furthermore, in this study, it was found that grounding of the toes can compensate for standing instability. But if the toes cannot be grounded even in an unstable condition, it may lead to falls. Therefore, we would like to conduct further research on the grounding of the toes in unstable states.

Comment 2:

Please correct the explanation of COP-path and COP-area so that the reader can understand it intuitively by using the example.

Also, please add the reason why the measurement time of 20 seconds was decided.

The reason why the shorter COP-path was used for the analysis in the two experiments is also unclear as it is.

Furthermore, I don't know if two experiments are enough.

Please add descriptions.

Response:

Thank you for your suggestion. The other reviewer raised a similar comment. The Method section has been revised to make it easier to understand COP. (Line 91-99)

Some of the literature that measured postural stability using force plate discuss the results using the average of the two-times measurements, but most of the literature uses the one with the better results (the smaller value of postural stability). In particular, in the case of children, I think it makes more sense to use the one with less sway because the variation of postural stability in each measurement is larger due to the effects of concentration and other factors. In addition, although reports on postural stability in adults typically use a 30-second test, we thought that a 30-second test is too difficult for 8-year-old children, and we adopted a 20-second test. There was a significant trend in the current study data, and we believe that the 20-second test is reasonable for 8-year-old children.

Comment 3:

I can understand the definition of the explanation of FT score used in this study, but I do not understand its specific meaning, for example, why the thumb touches the ground at 2 points.

Please indicate the reason for deriving this definition.

In addition, it is said that there are other FT evaluation indexes, so please explain the relationship with them and the original points that are significantly different.

In the text, there are some descriptions such as FT rate and FTS.

Make sure it's different or that it's the same thing with a different name.

Response:

Thank you for your suggestion. I had a similar comment from the other reviewer. 

I have revised the test procedure and protocol section. (Line 101-110)

FTS is a mistake for the FT score and has been corrected.

FT rate has been corrected to rate of FT.

Comment 4:

In the discussion, you said that the greater correlation between the FT score and the COP-path or COP-area when the eyes are closed is a result of the toes performing a function to stabilize the body.

But can't we interpret that instability is the cause of the toes' contact with the ground?

I think this is an important point, so please add a description.

Response:

Thank you for your comment. Although it is very difficult to prove the point, adults generally have less floating toe and more stable postural stability than children. Therefore, it is unlikely that the toes' contact with the ground is the cause of postural instability, and it is more reasonable to think that the toes are grounded to compensate for instability.

Comment 5:

"prevalence of floating toe" is included in the title.

Did you get any new insights into this?

In the discussion, it was written as the same as the previous reports, and I could not find a description of what was newly discovered.

Response:

Thank you for your comment. Although the result of the high frequency of floating toe in children is similar to previous reports, there are very few reports on floating toe in children. Our results confirmed the high frequency of floating toe in children. In addition, the result that the frequency of floating toe decreases in eye closed condition is a new finding. Thus, we chose this title based on these points. 

Comment 6:

It is very interesting that you are planning multifaceted research in the future, so please tell us specifically what kind of research results you can expect.

Response:

Thank you for your comment. We cannot discuss the future plan here, but we are considering investigating the relationship of postural stability and floating toe with several motor functions and exercise habits, and mental development. We also have plans to follow up with the same participants in the future to examine changes in floating toe and postural stability with age in the same cases. Although these points are described in the discussion section, I have slightly modified them as a future plan. (Lines 209-215)

---

## [Decision Letter · Decision Letter 1]

12 Mar 2021

Prevalence of floating toe and its relationship with static postural stability in children: The Yamanashi adjunct study of the Japan Environment and Children’s Study (JECS-Y)

PONE-D-21-00839R1

Dear Dr. Wako,

We’re pleased to inform you that your manuscript has been judged scientifically suitable for publication and will be formally accepted for publication once it meets all outstanding technical requirements.

Kind regards,

Tomoyoshi Komiyama, Ph.D

Academic Editor

PLOS ONE

Additional Editor Comments (optional):

Dear author,

Thank you for submitting your revised manuscript.

I think it was much easier to understand than the original manuscript.

I am satisfied with the responses and the edits, I am happy to accept this manuscript.

The authors have replied to my remaining comments satisfactorily from two reviewers.

Therefore, I have no further comments to make, all of my previous concerns were adequately addressed.

This manuscript will be satiating the reader's interest.

Tomoyoshi Komiyama

Reviewers' comments:

Reviewer's Responses to Questions

**Comments to the Author**

1. If the authors have adequately addressed your comments raised in a previous round of review and you feel that this manuscript is now acceptable for publication, you may indicate that here to bypass the “Comments to the Author” section, enter your conflict of interest statement in the “Confidential to Editor” section, and submit your "Accept" recommendation.

Reviewer #1: All comments have been addressed

Reviewer #2: (No Response)

2. Is the manuscript technically sound, and do the data support the conclusions?

Reviewer #1: Yes

Reviewer #2: Yes

3. Has the statistical analysis been performed appropriately and rigorously? 

Reviewer #1: Yes

Reviewer #2: Yes

4. Have the authors made all data underlying the findings in their manuscript fully available?

Reviewer #1: Yes

Reviewer #2: Yes

5. Is the manuscript presented in an intelligible fashion and written in standard English?

Reviewer #1: Yes

Reviewer #2: Yes

6. Review Comments to the Author

Reviewer #1: The authors properly reflect the findings. By classifying men and women in Tables 3 and 4, the gender difference can be shown more clearly. Initially, the consideration of this difference was "difference in brain structure between men and women", but it has been corrected by appropriately.

Reviewer #2: I think you have responded appropriately to my comments in the previous peer review. Thank you very much.

7. PLOS authors have the option to publish the peer review history of their article (what does this mean?). If published, this will include your full peer review and any attached files.

Reviewer #1: No

Reviewer #2: **Yes: **NORIO TAGAWA

---

## [Editor Report · Acceptance letter]

16 Mar 2021

PONE-D-21-00839R1 

Prevalence of floating toe and its relationship with static postural stability in children: The Yamanashi adjunct study of the Japan Environment and Children’s Study (JECS-Y) 

Dear Dr. Wako:

I'm pleased to inform you that your manuscript has been deemed suitable for publication in PLOS ONE. Congratulations! Your manuscript is now with our production department. 

Kind regards, 

on behalf of

Dr. Tomoyoshi Komiyama 

Academic Editor

PLOS ONE